# Rapid realist review of the role of community pharmacy in the public health response to COVID-19

Ian Maidment [ID] ,[1] Emma Young,[2] Maura MacPhee,[3] Andrew Booth,[4] Hadar Zaman,[5] Juanita Breen,[6] Andrea Hilton,[7] Tony Kelly,[8] Geoff Wong[9]

For numbered affiliations see end of article.

**Correspondence to**
Dr Ian Maidment;
i.maidment@aston.ac.uk

## ABSTRACT

**Introduction** Community pharmacists and their teams have remained accessible to the public providing essential services despite immense pressures during the COVID-19 pandemic. They have successfully expanded the influenza vaccination programme and are now supporting the delivery of the COVID-19 vaccination roll-out.

**Aim** This rapid realist review aims to understand how community pharmacy can most effectively deliver essential and advanced services, with a focus on vaccination, during the pandemic and in the future.

**Method** An embryonic programme theory was generated using four diverse and complementary documents along with the expertise of the project team. Academic databases, preprint services and grey literature were searched and screened for documents meeting our inclusion criteria. The data were extracted from 103 documents to develop and refine a programme theory using a realist logic of analysis. Our analysis generated 13 context-mechanism-outcome configurations explaining when, why and how community pharmacy can support public health vaccination campaigns, maintain essential services during pandemics and capitalise on opportunities for expanded, sustainable public health service roles. The views of stakeholders including pharmacy users, pharmacists, pharmacy teams and other healthcare professionals were sought throughout to refine the 13 explanatory configurations.

**Results** The 13 context-mechanism-outcome configurations are organised according to decision makers, community pharmacy teams and community pharmacy users as key actors. Review findings include: supporting a clear role for community pharmacies in public health; clarifying pharmacists' legal and professional liabilities; involving pharmacy teams in service specification design; providing suitable guidance, adequate compensation and resources; and leveraging accessible, convenient locations of community pharmacy.

**Discussion** Community pharmacy has been able to offer key services during the pandemic. Decision makers must endorse, articulate and support a clear public health role for community pharmacy. We provide key recommendations for decision makers to optimise such a role during these unprecedented times and in the future.

### Strengths and limitations of this study

► A diverse group of professional and public stakeholders validated our findings from the literature.
► By using a realist approach, we were able to use a broad range of data, including grey literature.
► To enable us to develop recommendations in a timely manner the focus of the review was deliberately narrow.
► The topic could have been informed by other sources of evidence in particular empirical interviews with key stakeholders.
► COVID-19 vaccination is a rapidly evolving area and this research was based on the best available evidence at the time.

## INTRODUCTION

Community pharmacy teams have continued to provide essential services during the COVID-19 pandemic. They offer accessibility and medicines expertise to the public, even in challenging times.[1–3] However, COVID-19 creates extra workload demands, such as medication dispensing with increases of up to 33% in prescription numbers.[4] To cope with this demand community pharmacies have increased their opening hours and hired additional staff.[4] Alongside this additional workload, they have managed widening coverage of the influenza vaccines programme.[5]

Evidence suggests that community pharmacy can successfully provide diverse vaccination services including seasonal and pandemic influenza, travel vaccinations and hepatitis B for at-risk groups, within the provisions of the UK National Health Service (NHS) or privately.[6] They have successfully provided influenza vaccines as an NHS commissioned advanced service since 2015.[5] One service evaluation found that of 485 patients asked, 99% expressed confidence in their pharmacist to provide additional vaccinations.[7] Community pharmacy can also support influenza and other vaccinations to combat the

BMJ

significantly higher COVID-19-related mortality in ethnic minorities (excluding White minorities).[8 9]

The COVID-19 pandemic has stretched NHS capacity to safely and efficiently meet public health demands. A role for community pharmacy in the national vaccination service requires an understanding of what pharmacy teams require to successfully deliver essential and advanced services during the pandemic. Such knowledge is timely, given the roll-out of COVID-19 vaccines across the community.[10]

Delivering a vaccination programme is a complex process and successful delivery is context dependent. A realist review helps make sense of complex situations,[11 12] such as how community pharmacy can most effectively address the challenges presented by COVID-19. A rapid review can generate guidance for decision makers to assist with roll-out of COVID-19 vaccinations to community pharmacy. This rapid realist review aimed to understand how community pharmacy can most effectively deliver essential and advanced services, with a focus on vaccination, during the pandemic and in the future.

## METHODS

A rapid realist review of academic and other literature, supplemented by input from key actors, was undertaken to understand how and when community pharmacy can effectively support the public health agenda during pandemics such as COVID-19. Rapid reviews aim to ensure findings are generated and disseminated in response to the urgent nature of the situation. To produce this knowledge at pace, we truncated the following review processes:

► Programme theory development was undertaken within 1 month with input from the project team.
► Searching was expedited using broad search terms and using a limited number of key data sources.
► Data analysis and context-mechanism-outcome configuration (CMOC) development focused on where the programme theory was considered most important during COVID-19.

This realist review was undertaken within 6 months (August 2020–January 2021), the protocol was published on PROSPERO[13] and, where relevant, follows the Realist And Meta-narrative Evidence Syntheses: Evolving Standards (RAMESES) quality and publication standards.[14]

### Stage 1: programme theory development

The project team met virtually to develop an embryonic programme theory using four diverse documents from an initial search representing a professional journal,[15] a research journal,[16] a policy document (Royal Pharmaceutical Society (RPS))[17] and a practical influenza briefing.[18] The team identified the need for (1) enabling guidance for community pharmacy (to achieve legitimisation)[16]; (2) practical direction for community pharmacy practices (to ensure feasibility)[15 16]; and (3) user assurance of appropriate, safe, feasible and timely intervention (relative advantage).[15] The resultant embryonic theory,

patterned on a COM-B behavioural model of capability, opportunity and motivation leading to behaviour,[19] was used to inform searching and initial analyses.

### Stage 2: literature searching

Searches were conducted (July–August 2020) using MEDLINE, EMBASE, CINAHL, Web of Science and Scopus for search concepts relating to Pharmacy and COVID by AB (see online supplemental appendix 1 for search strategy). Reference checking and citation searching of all included references on Google Scholar (using the Publish or Perish tool) were also carried out. Given the novelty of the virus, we searched the contents of preprint services and the WHO COVID Register. Grey literature searches included social media (eg, blogs, facilitated Twitter discussion (#Cpharmchat)), community pharmacy websites and emails from relevant regulators and professional organisations (eg, RPS, Pharmaceutical Services Negotiation Committee, General Pharmaceutical Council).

Key inclusion criteria were high or middle-income countries, community pharmacy and infectious disease management (see online supplemental appendix 2). The search covered January 2003–July 2020 to include SARS, a comparable condition first identified in 2003. There were no restrictions on study designs eligible for inclusion.

### Stage 3: data selection and extraction (selection and coding)

Selection and appraisal of documents followed a two-step procedure:

1. EY screened the title, abstract and keywords of potentially relevant documents against inclusion criteria. A 10% random sample was checked by two research team members (AB and MM) for consistency.
2. EY obtained and screened full texts of all documents meeting the eligibility criteria.

Relevant data from the included full-text documents were coded into NVivo by EY, MM and JB. Some codes came from the data (ie, inductive coding); others were derived from the programme theory (ie, deductive coding) and some were derived using retroduction (ie, by interpretation of what might be functioning as mechanisms).[20] No assessment was made of the rigour of the data within included documents; however, global judgements were made of the quality of the explanations provided by the CMOCs and programme theory using the criteria of consilience, simplicity and analogy.[21]

### Stage 4: data synthesis

The data analysis/synthesis was conducted by EY, MM and JB with input from the rest of the project team to develop and refine the programme theory using a realist logic of analysis. Our analysis generated 13 realist CMOCs, explaining when, why and how community pharmacy can support public health vaccination campaigns, maintain essential services during pandemics and capitalise on opportunities for expanded, sustainable public health service roles. Actor conversations generated further

CMOCs related to care for diverse and vulnerable populations, including ethnic minorities.

Our realist logic of analysis centred on the following questions:

► Interpretation of meaning: do the contents coded by the team provide data that may be interpreted as CMOCs?
► Interpretations and judgements about CMOCs: how do the CMOCs relate to the programme theory?
► Interpretations and judgements about programme theory: how do the programme theory and its CMOCs correspond with key actor perspectives of reality?

Data to answer our questions were iteratively sought across documents. Interpretive cross-case comparison was used to identify and explain the 'success' of pandemic community pharmacy interventions delivered in different settings or to different population groups.

### Key stakeholders

Key stakeholders, including community pharmacists and support staff (including representatives from large and smaller chains, sole independent pharmacies and primary care), other healthcare professionals and members of the public were consulted on four occasions. The meetings took place over Microsoft Teams and each lasted about 1 hour.

### Patient and public involvement

Members of the public were drawn from The University of Sheffield's Patient and Public Involvement database and contacts of the core project team. Groups numbered between 11 and 13 members, with ages ranging from 22 to 74 years, from diverse locations and ethnicities including Black African, Black Caribbean, British Asian, British Chinese, White Irish and White British. Collectively, stakeholders provided feedback and advice on their real-world experience of working in or using community pharmacy.

## RESULTS

One hundred and three documents were included in this rapid review and were coded to develop and refine our CMOCs and programme theory (Preferred Reporting Items for Systematic Reviews and Meta-Analyses diagram in online supplemental appendix 3). The final programme theory is summarised in online supplemental appendix 4—from abstract visioning to actual patient uptake of the COVID-19 vaccine. Although the programme theory is outlined in a linear fashion, steps within it are not necessarily linear and may occur simultaneously. The CMOCs are organised according to key actors, or individuals and groups with a vested interest in community pharmacy delivery of a COVID-19 vaccination programme. 'Actor' derives from sociology and is synonymous with 'stakeholder'; we privileged this term to differentiate programme theory/CMOC organisation from project stakeholder participants.[22] Table 1 briefly describes the three actor groups with their respective CMOCs and corresponding steps in online supplemental appendix 4. Tables 2–4 show the final 13 CMOCs.

The following sections summarise the CMOCs related to each of the three actor groups. Illustrative examples of the supporting evidence from review documents are presented (tables 2–4).

### Decision makers

Decision maker and public endorsement (CMOC 1) are essential first steps in enlisting community pharmacy for COVID-19 vaccination programmes. Regulators must ensure pharmacists have the legal scope to do so (CMOC

**Table 1** Programme theory actors with corresponding CMOCs and steps

| Programme theory actors | Description | CMOCs* | Steps in programme theory† |
|---|---|---|---|
| Decision makers | The UK government, regulatory and professional bodies, the public. | 1–4 | 1–4 |
| Community pharmacists and their teams | Community pharmacists are healthcare professionals registered by the General Pharmaceutical Council and supported by teams made up of counter assistants, dispensers and registered technicians. They work in high street locations, in local communities and in supermarkets. Employers range from large chains to small individually owned community pharmacies. | 5–9 | 5–7 and 9–11 |
| Pharmacy users | Members of the public who use any community pharmacy services including prescription dispensing, minor ailment advice/treatment or vaccination services. | 10–13 | 8 |

*See tables 2–4.
†See online supplemental appendix 4.
CMOC, context-mechanism-outcome configuration.

**Table 2** Decision makers (CMOCs 1–4)

| | |
|---|---|
| CMOC 1— support a public health role[23–27 62 66–69] | **When the government, pharmacy regulators, professional bodies and the public endorse and support a clear role for community pharmacy in public health services (C), community pharmacists will be more likely to adopt vaccination services (O) because they see it as their professional role and duty (M).** <br><br> *'Distribution and administration of the COVID-19 vaccination programme will require concerted action across the NHS. With unique insight and expertise in medicines and the delivery of vaccination programmes, pharmacists have a clear role in contributing to the success of this programme.' (Great Britain)*[66] <br><br> *'55% of the public have visited a pharmacy during the COVID-19 crisis…89% of people believe pharmacies are playing an essential role in the COVID-19 crisis.' (UK National Pharmacy Association)*[27] <br><br> *'Pharmacists…have been called on to coordinate the administration of COVID-19 tests…providing ongoing COVID-19 surveillance to communities by allowing walk-in testing at community pharmacies… [This] might be more sustainable and convenient than the large-scale public screening being done as of the summer of 2020.' (USA)*[23] <br><br> *'Given the past success of community pharmacists with increasing annual seasonal influenza uptake and their accessibility, pharmacists will need to be central in administering COVID-19 vaccines in order to achieve rapid population-wide coverage.' (Canada)*[26] |
| CMOC 2—clarify legal and professional liabilities[24 28 66 70–76] | **When pharmacy regulators and the NHS clarify community pharmacists' legal and professional liabilities arising from the administration of a novel and potentially unlicensed COVID-19 vaccine (C), community pharmacists are more willing to give the vaccination (O) because they feel reassured regarding liability (M).** <br><br> *'The role of pharmacists in the COVID-19 vaccination programme must be made clear to the pharmacy profession itself. Professional and representative pharmacy bodies have an important role to play in providing the right level of information to the profession to support their roles in the vaccination programme.' (Great Britain)*[66] <br><br> *'Indemnity insurance for individual healthcare professionals needs to be amended to cover this activity and be state-funded. There also needs to be clear communication to healthcare professionals, so they clearly understand that they are covered and under which circumstances this applies.' (Great Britain)*[77] |
| CMOC 3—Codevelop feasible service specifications[1 28 37 54 78–82] | **When COVID-19 vaccination policy and service specifications have been developed with input from diverse community pharmacists and staff tasked with administering and supporting the administration of the vaccine (C), community pharmacies are more likely to deliver the service (O) because they believe the service specification is feasible (M).** <br><br> *'Pharmacists ideally want input into future policy changes before they are finalized, so that these can reflect capacity and preparedness on the ground and be publicized accurately.'* (Great Britain)[1] |
| CMOC 4—issue clear, relevant and timely guidance[23 24 28 31 35 37 46 67 71 83–92] | **When government, pharmacy regulators and professional bodies provide consistent, clear, relevant and timely guidance for the delivery of the COVID-19 vaccines (C), community pharmacies are more likely to deliver the service (O), because the guidance is helpful and simplifies implementation (M).** <br><br> *'… it is good to see that NHSE/I have provided the information we have been waiting for to review the resources we have nationwide and decide how we can bring them to bear to help the NHS defeat this virus.'* (Great Britain)[28] |

CMOC, context-mechanism-outcome configuration; NHS, National Health Service; NHSE, National Health Service England.

2), with community pharmacy input during the development of policies and protocols (CMOC 3), so that final service specifications are flexible and doable within local settings (CMOC 4).

In England community pharmacies have government contracts and partnerships to deliver vaccinations and other essential services during emergencies, including the COVID-19 pandemic.[23–25] Pharmacies in the USA and Canada have also been identified as having a substantive role in vaccine administration (CMOC 1).[23 26] The idea of harnessing UK community pharmacy capacity enjoys widespread public support (CMOC 2).[27] However, appropriate service delivery is hampered by unfeasible operational conditions (CMOC 3)[1]; for example, medication deliveries are funded by the UK government for 'vulnerable people', but this category of service users is defined narrowly by the government and misunderstood by the public, creating unrealistic expectations of community pharmacy and generating additional work.[1]

**Table 3** Community pharmacists and team (CMOCs 5–9)

| CMOC 5—receive adequate compensation and resources[1 30 32 40 45 93–103] | **When community pharmacies receive adequate compensation and resource support for COVID-19 vaccines (C), they are more likely to deliver the service (O), because they feel recognised for their service contributions (M).** |
| --- | --- |
| | 'Pharmacists…have been incredible in supporting patients throughout COVID-19 and rightly deserve recognition for the work they do. We know that teams were already under pressure and that colleagues in community pharmacy are feeling added financial strain.' (Great Britain).[30] |
| | 'In the community, mitigating the impact of COVID-19 has…focused largely on general practitioners (GPs). This is unsurprising to those in the pharmacy profession who have long considered policy makers to overlook them.' (Great Britain)[1] |
| CMOC 6—sustain capacity and facility to adapt essential services[16 17 32–34 69 71 96 97 104–115] | **When a community pharmacy has the capacity and permission to manage and adapt existing essential services during COVID-19 (C), they are more likely to effectively deliver and sustain these services (O) because it is feasible for them to do so (M).** |
| | 'Virtual and telephone consultations have become commonplace, particularly to vulnerable patients. Pharmacists have implemented systems to dispense medications in advance of need to minimise wait times and duplicate visits. In case-by-case examples…there has been anticipatory management of medication-related needs.' (Ireland)[115] |
| | 'Pharmacists who had comfort and confidence in managing electronic communication reported feeling greater control over workflow and the ability to triage and queue patients more effectively based on priority and need.' (Canada)[46] |
| CMOC 7—inform users of essential service availability/continuity[16 24 25 33–35 46 62 97 109 115–124] | **When community pharmacies have processes in place to inform pharmacy users about the availability of essential services (C), pharmacy users are less likely to become anxious (O), because they feel reassured about access to what they need (M).** |
| | 'Access to medications is a main concern expressed by our patients… The CDC and SAMHSA offer guidance on ways to reduce stress and anxiety in this time of uncertainty.'[118] |
| | 'As a result of the power dynamics at play, it is ultimately up to pharmacists to be able to reassure patients and provide care, all while taking into account their mental health. Currently, guidelines regarding patient interaction during a pandemic are needed.'[35] |
| | 'In addition, there is ongoing work in liaison with pharmacists, general practitioners, and state and territory authorities to enable therapeutic substitution by pharmacists in the event of a shortage. This will allow community pharmacists to substitute dose strength or form without prior approval from the prescriber, if a prescribed medicine is not available at the time of dispensing. These measures highlight the important role pharmacists can play in enabling and maintaining access to medicines for people in need throughout the COVID-19 outbreak.' (Australia)[109] |
| | 'Access to medications is a main concern expressed by our patients… The CDC and SAMHSA offer guidance on ways to reduce stress and anxiety in this time of uncertainty.' (USA)[118] |

Continued

**Table 3** Continued

| CMOC 8—protect the health and safety of staff[24 34 35 55 66 97 103 125–130] | When community pharmacies have the means to protect the health of pharmacists and staff (C), they are more likely to deliver essential services and COVID-19 vaccination services (O), because they feel safe to do so (M). |
| --- | --- |
| | '…Given high levels of public anxiety and uncertainty regarding the integrity of the drug supply chain in Canada, many participants reported difficult, sometimes frightening, interactions with members of the public and their desire for dedicated security to provide support and conflict management.' (Canada)[24] |
| | 'An additional burden Asian pharmacists face, on top of pharmacist harassment, is the rise of anti-Asian racism that has come about due to COVID-19. Verbal and even physical abuse has been reported to happen in various countries, such as the UK, France, and the USA, to those of Chinese descent…' (UK, France, USA)[35] |
| | 'The survey found that although more than a third (37%) of pharmacists said they felt unsafe at some point working during the pandemic, concerns over PPE have eased. This is due, in part, to the supply problems being eased and community pharmacists being finally allowed to order from the national online PPE portal in an emergency, after calls for access from the RPS.' (Great Britain)[103] |
| CMOC 9—enhance collaboration across services, including IT[18 26 33 40 41 82 88 89 104 109 131–135] | When systems needed for COVID-19 vaccination, including IT and remuneration, support collaboration by various service providers (eg, general practitioner surgeries and community pharmacies: (C)), a coordinated response is more likely (O), because reduced effort is needed from everyone (M). |
| | 'Planning and delivery should be undertaken across a consistent, pre-agreed footprint. It may be more efficient and cost effective to provide immunisation across a number of providers, pooling resources and sites to deliver the best service possible, and working in coordination with other local stakeholders such as directors of public health and local government.' (Great Britain)[82] |
| | 'For many community pharmacists, a lack of connected IT is a huge problem. Kieran Eason, who runs an independent pharmacy in Tamworth, Staffordshire, says lack of intra-operability makes it more difficult to do relatively simple things, like sending prescription requests to GPs. "Pharmacy IT is just a complete disaster," he says, suggesting the COVID-19 crisis has highlighted flaws, such as the number of different systems pharmacists use.' (Great Britain)[40] |

CDC, Centers for Disease Control and Prevention; CMOC, context-mechanism-outcome configuration; IT, information technology; PPE, personal protective equipment; RPS, Royal Pharmaceutical Society; SAMHSA, Substance Abuse and Mental Health Services Administration.

**Table 4** Pharmacy users (CMOCs 10–13)

| | |
|---|---|
| CMOC 10—trust the pharmacist as reliable information source[25 61 85 136–138] | **When pharmacy users receive information about the COVID-19 vaccine from their community pharmacy (C), they are more likely to get the vaccine (O), because they trust community pharmacy as a source of reliable, accurate information (M).** |
| | 'When patients were educated about influenza, herpes zoster, and pneumococcal vaccines as a result of a pharmacist-driven intervention in community pharmacies, they were influenced to receive the vaccination.' (USA)[136] |
| | 'Being able to address the public enquiries with accurate up-to-date information about the local situation and the overall infection progress is the key to build trustful relationship with them at troubled times.' (Macau)[25] |
| CMOC 11—trust the pharmacist to deliver responsive services[7 18 26 27 30 31 35 42 43 54 71 74 86 96 108 109 139–149] | **When community pharmacies are trusted to make necessary service adaptations to ensure services are flexible, convenient and accessible (C), COVID-19 vaccine uptake is likely to be higher among pharmacy users (O) because the service is responsive to local needs (M).** |
| | 'Pharmacists have always been the most accessible health care provider; this is especially true in the era of COVID-19….While other professionals have closed their doors to patients, community pharmacies remained open to the public despite stricter lockdown restrictions. As highly trusted healthcare clinicians, community pharmacists play a vital role in closing the gaps that are exacerbated by the additional strain on the system and reduced access to healthcare providers.' (Canada)[35] |
| | 'Pharmacy flu vaccination services complement those provided by general practitioners to help improve overall coverage and vaccination rates for patients in at-risk groups. These services are highly accessed by patients from all socio demographic areas, and seem to be particularly attractive to carers, frontline healthcare workers, and those of working age.' (Great Britain)[149] |
| CMOC 12—access culturally sensitive services[33 67 104 115 150] | **When community pharmacies leverage their community location and community–staff relationships (C), vulnerable populations, such as ethnic minorities, are more likely to use their services, including COVID-19 vaccination services (O), because they trust their local community pharmacies to provide culturally sensitive service (M).** |
| | 'Research continues to highlight that patients who are medically under-served have poorer inequitable access to health care due to them experiencing greater physical barriers to accessibility, encountering poorer patient-professional communication and are significantly disadvantaged where a service is not tailored to their unique needs or preferences.' (Great Britain)[151] |
| | 'I think over the years what's happened is nationally it's almost like everything has to be the same, which then doesn't work because it doesn't accommodate all the little variations…So we need to go back and in each individual pharmacy, gear it towards the population that it is meant to be meeting the needs of.' (Pharmacist caring for ethnic minority community—Great Britain)[151] |
| CMOC 13—receive private and confidential services[7 43 45 83 152–154] | **When community pharmacies make provisions for privacy (C), pharmacy users are more likely to use their services (O), because they are reassured about confidentiality (M).** |
| | 'If there was something not right…the first thing I would do is make an appointment with a doctor. I wouldn't do and talk to somebody over a pharmacy counter.' (Great Britain)[45] |
| | 'Privacy, confidentiality and dignity are all vital elements of a trusting relationship between healthcare professionals and their patients…. In terms of privacy, the quality was perceived by the participants to include a confidential room that enabled private consultations.' (Great Britain)[7] |

CMOC, context-mechanism-outcome configuration.

In contrast, when clear options for community pharmacy involvement in COVID-19 vaccination programmes were issued through the NHS (CMOC 4), the chief executive of the Association of Independent Multiple Pharmacies commented positively on members' engagement in delivering vaccines.[28]

### Community pharmacists and team

Community pharmacies have had to manage ongoing essential services, in addition to supporting COVID-19 vaccination delivery during the pandemic (CMOC 6), including delivery of necessary medications (CMOC 7). Given fears and anxieties related to COVID-19 and changes to service delivery, community pharmacies have had to deal with inappropriate behaviours from the public, including emotional abuse and threats of physical abuse (CMOC 8). As an integral public health service, community pharmacy capacity to meet NHS needs will be enhanced through use of information technology (IT) and collaboration with other service providers (CMOC 9).

Although attention is focused on a COVID-19 vaccine, community pharmacies offer diverse essential services. In the UK and other high-income countries, community pharmacy services also provide advanced services such as vaccinations.[29] Pharmacists consider it a professional

responsibility to provide essential services during the pandemic, despite clear financial risks to themselves.[30]

Despite pharmacists' professional and moral obligations to provide essential services, ongoing persistent 'under-recognition' can jeopardise their ability to contribute to COVID-19 vaccinations (CMOC 5) as well as maintain a usual service. Under-recognition has been an issue in previous UK vaccination campaigns; respondents to a survey of Welsh community pharmacists after the 2016 influenza season described providing a 'mop up' service for general practitioners (GPs).[31 32]

Multiple required community pharmacy service adaptations have been reported by the UK Pharmaceutical Journal, including change-over in retail space to medication preparation and dispensing, and call-in shopping services for other retail items (CMOC 6). Service adaptations have been affected by hours of operation, available staff and cancellations of contracted services, such as blood pressure testing and smoking cessation support—all with financial implications for pharmacists.[33]

Before COVID-19, community pharmacies globally were offering 'valued-added services' (VAS) such as drive-thru services, online ordering and communications services (eg, prescription reminders) to stay competitive. Many VAS have helped pharmacies adapt more quickly to pandemic restrictions.[34]

During the pandemic, ensuring access to needed medications has been a critical pharmacy service, to allay public concerns (CMOC 7).[16] Potential and actual disruptions in expected services and needed supplies (eg, medications) have resulted in tensions, threats and verbal/physical abuse by the public to community pharmacists (CMOC 8).[33 35 36] Safety policies, protocols and safety-related supplies (eg, personal protective equipment (PPE)) must be in place to ensure community pharmacy teams' safety.[33]

The UK COVID-19 vaccination campaign borrows heavily from previous, successful collaborative influenza vaccination programmes using community pharmacy and GPs (CMOC 9).[37] Pooling resources improves service delivery.[38] For example, the UK Pharmaceutical Journal reported how a greater collaborative approach has resulted in successful influenza vaccination of home care staff and domiciliary workers during the pandemic.[38]

The pandemic has clearly demonstrated the importance of interoperable, connected IT systems across services.[39] Getting access to reliable information is important for tracking supplies and deliveries related to the COVID-19 vaccination programme.[40] A pandemic silver lining is raised awareness of IT functions for enhanced delivery of essential services (eg, medication planning, prescribing and dispensing between pharmacies and GPs) and advanced services, including vaccinations.[40]

### Pharmacy users

Pharmacy users trust community pharmacies as a reliable source of information (CMOC 10) about vaccines, and pharmacies' local accessibility and convenience increases the likelihood of users obtaining COVID-19 vaccines through them (CMOC 11). Community pharmacy relationships with vulnerable populations in their local settings may enhance uptake of the vaccine by these groups (CMOC 12). Provision for privacy is an important user consideration. Pharmacy users expect private consultations to preserve their confidentiality (CMOC 13).

Public trust in community pharmacists is high—similar to doctors and nurses.[28] Trust between pharmacists and users can be leveraged to overcome scepticism about the COVID-19 vaccine. Providing reliable information about the disease and the vaccine, as pharmacies have done with other infections, can enhance public uptake of the COVID-19 vaccine.[41] In the USA, 90% of the US population lives within 5 miles of a community pharmacy. Given their convenience and accessibility, consumers have visited their community pharmacists 12 times more frequently than their GPs.[21] A UK study found that consumers who were eligible for a free influenza vaccine through their GPs were willing to pay for pharmacy service because of convenience and ease of access.[42] Established, trusting relationships are especially important for providing culturally sensitive services to marginalised and vulnerable communities.[43] Community pharmacists, often members of local communities, are specially positioned to understand the culturally contextual factors that impact their pharmacy users.[44] Trusting relationships are founded on privacy, confidentiality and dignity, and COVID-19, public health protocols and limited space must be considered—in order to maintain vital trust among pharmacy users.[45]

### DISCUSSION

This realist review sought to understand how community pharmacy can contribute to the public health agenda during the COVID-19 pandemic, particularly continuation of essential services and engagement in vaccination services.[29] As the COVID-19 vaccination service continues to evolve, our recommendations for decision makers highlight opportunities for community pharmacy to promote safe, efficient and effective service delivery.

### Summary of key findings

To optimise community pharmacy service during the pandemic, decision makers must endorse and articulate a clear role for these healthcare professionals. The public already endorses advanced roles for community pharmacy (eg, vaccinations, minor ailment scheme), but public awareness depends on what decision makers do and say. Practical decision maker measures include adequate reimbursement to help cover the cost for time,

staff and PPE (particularly for a sustainable long-term service); legal (including indemnity), regulatory coverage for advanced roles; and clear and consistent guidance for vaccination preparation and for adaptation of essential services. When given the opportunity, the permissions and resources to do so, community pharmacies have been able to adapt quickly to continue essential services and whenever possible, to offer critical advanced services. Historically, community pharmacies have significantly increased vaccination uptake (eg, influenza vaccinations) given their accessibility and convenience and capacity to adapt to local needs for the general population and marginalised groups.

### Compare to other similar or related studies discussing important differences in results

At the time of this realist review, there were no similar reviews on community pharmacy roles with respect to COVID-19 vaccinations. COVID-19 represents an unprecedented situation with limited direct evidence to guide decision-making. However, realist approaches engage with a wider evidence base, including research on past pandemics (eg, SARS),[46–48] mass vaccination campaigns (eg, influenza)[49] and community pharmacies' capacity pre-COVID-19 to deliver essential and advanced services.[19 50 51] Key factors previously include a lack of leadership, a lack of guidance and an increasing reliance on professional judgement and experience.[46] Research from the UK and other economically developed countries supports the 13 CMOCs and decision maker recommendations in this review (see tables 2–5). However, limitations continue to surround the direct policy relevance of much of the community pharmacy evidence base.[52]

### Strengths and limitations

The realist approach uses diverse data, including grey literature. This feature is especially important given a novel and rapidly evolving topic area, such as COVID-19. Multiple researchers with subject matter expertise participated in screening the literature and extracting and coding data, which maximised opportunities to discuss and debate the plausibility of the inferences made. The CMOCs were developed and refined through regular discussions within a team with varied academic and clinical backgrounds. Professional and public stakeholder consultation further refined the CMOCs.

All rapid reviews operationalise coverage versus expediency. Other sources of evidence could have informed the review; however, potential gaps were mitigated by stakeholder engagement and expertise within the team. The evidence supporting the CMOCs was based on available time for document review, and during the review period, research, directives and policy related to COVID-19 vaccination rapidly evolved. The programme theory and its CMOCs, however, are expressed in such a way that they can be further confirmed, refuted or refined in the future using additional data.

For this rapid review, given our short timeline of 6 months, we initially decided to focus on community pharmacy roles and pandemic response in middle to high-income countries. While we identified data from higher income countries such as Canada, USA, Australia and France, there was very little published from middle-income countries. As such, it is likely that our findings and recommendations are most applicable to high-income countries, although the lessons learnt from high-income countries can serve as an initial reference point for best practice for other countries.

**Table 5** Recommendations for decision makers to increase community pharmacy engagement in pandemic response (for further details check: https://publications.aston.ac.uk/id/eprint/42310/1/Guidance_for_Policy_Makers_on_the_role_of_Community_Pharmacy_in_COVID.pdf)

| Recommendations | Derived from CMOC |
|---|---|
| Articulate a clear public health agenda role for community pharmacy (eg, COVID-19 testing and vaccination). | 1 |
| Ensure pharmacy regulations for advanced roles, such as novel vaccine administration, are in place to legally protect community pharmacists and their teams. | 2 |
| Involve local community pharmacies in policy and service specification development. | 3 |
| Provide timely guidance with sufficient details for community pharmacies to quickly adapt to local needs. | 4 |
| Provide adequate funding and reimbursement for community pharmacy services to deliver COVID-19 vaccines. | 5 |
| Equip community pharmacies with the necessary permissions to manage and adapt essential services. | 6 |
| Ensure community pharmacies have the means to adequately protect the health of themselves, their staff and pharmacy users. | 8 |
| Facilitate collaboration and coordination of COVID-19 vaccination services across providers (eg, GPs, community pharmacies) and systems (eg, IT). | 9 |

CMOC, context-mechanism-outcome configuration; GP, general practitioner; IT, information technology.

Our recommendations can be viewed as 'how tos' for other countries to consider within their own cultural and healthcare environments. A number of European and North American countries have introduced legislative changes to extend the role of community pharmacy and thus reduce pressure on other parts of their healthcare systems.[51 53] Our findings can be used by community pharmacy and decision makers, from countries other than the UK, when mapping out similar services particularly in the context of changes in the legislation.

## Meaning of the study: possible explanations and implications for decision makers

Although there are multiple actors involved in pandemic response, for brevity, recommendations (table 5) are directed towards decision makers who possess the formal authority to implement the recommendations.

As the CMOCs were refined, 'tensions' were uncovered with implications for decision makers, particularly: community pharmacists as healthcare professionals versus retailers[51 54–56]; community pharmacies' capacity to remain financially viable while managing essential and advanced services[19 57 58]; and pharmacists' capacity to provide offsite services while maintaining physical premises.[59] These tensions stem from lack of awareness of community pharmacists as healthcare professionals.

Along the continuum of healthcare from community to hospital, community pharmacy is often the first point of contact for the public. Community pharmacies contribute to primary care services through essential services (eg, medication dispensing) and expanded roles (eg, vaccinations),[29 56] often decreasing workload pressures on other providers, such as GPs.[60] Public surveys demonstrate high levels of satisfaction with community pharmacy services, and vaccination uptake is increased when pharmacies deliver these services.[58] Nevertheless, lack of public and decision maker awareness of community pharmacy primary care roles has slowed uptake and integration of these services.[61] A recent UK public survey found that community pharmacies are seen as 'a medicine supply shop by 48.3% of people, as a place to purchase medicines by 22%, and a place to purchase non-medicinal products by 17.7%'.[62]

In Ireland and Canada, community pharmacies are an integral part of national vaccination campaigns.[58] In both instances, community pharmacists participate in vaccination planning, pharmacy regulators provide clear guidance on vaccination management and vaccinations are equitably refunded through public health systems. The removal of barriers, such as economic pressures on pharmacies, has resulted in impressive national vaccine uptake, even in large countries, such as Canada.[58] Globally, there is an emerging trend for governments, health insurance companies and consumers to remunerate community pharmacies for services that contribute to improved health outcomes.[59]

Community pharmacies tend to be defined as premises where medications are dispensed, which compounds the confusion (by the public and decision makers) of professional primary care services versus retailers.[59] Instead, pharmacies should be defined with respect to actions or services that require specialised health knowledge to optimise health outcomes. For example, a retail approach to over-the-counter (OTC) medications is to permit consumers to make their own choices, similar to supermarket choices. Pharmacy input into OTC purchases could potentially decrease adverse medication interactions or unnecessary allergic reactions.[56] The 'value-add' of community pharmacies, evidence-informed engagement with consumers, can decrease morbidity and mortality outcomes and increase medication regimen adherence.[59 63]

Although community pharmacists can provide professional services off premises (eg, immunisations in community and religious centres), they need to maintain their physical premises and staff for financial reasons. In addition, pharmacy users seek out community pharmacy services due to their accessibility and convenience, and without a physical space to engage regularly with pharmacy users, trust building between pharmacists and users is compromised.[64]

As evidenced by public media, many countries, including the UK, are declaring service specifications for COVID-19 vaccinations. Community pharmacists can potentially provide vaccination services in two locations.[65] They can collaborate with GPs in Primary Care Networks (PCN) to support PCN vaccination sites. Alternatively, they can provide a COVID-19 vaccination service from their premises if they meet service specifications. The described tensions create difficult choices for pharmacists that can be ameliorated through the decision maker recommendations in table 5.

## Unanswered questions and future research

A future realist evaluation involving primary data collection from key actors will inform refinement of the programme theory, CMOCs and decision maker recommendations. This empirical research will address the tensions identified above. Future research will also address issues identified through the stakeholder groups, such as vaccination hesitancy, outside the scope of this review. While this work touched on unique issues for marginalised populations including ethnic minorities, those of lower socioeconomic status and those with disabilities, further exploration is needed.

We had initially focused the review on community pharmacy roles and pandemic response in middle to high-income countries, but found limited data in middle-income countries. The findings are thus more likely to be applicable to high-income countries but may serve to inform practice elsewhere. Empirical research using, for example, a realist evaluation approach should be conducted to extend our findings on the role of community pharmacy in COVID-19 vaccination programmes to middle to low-income countries.

## CONCLUSION

The COVID-19 pandemic is a worldwide health emergency. Vaccination is key to combating the pandemic. The role of community pharmacy may be both short term and long term; with a potential need for regular annual vaccines. This rapid realist review offers recommendations for decision makers to enable community pharmacy to play a key role, both during these unprecedented times and into the future.

**Author affiliations**

[1]College of Health and Life Sciences, Aston University, Birmingham, UK
[2]The University of Sheffield, Sheffield, UK
[3]The University of British Columbia, Vancouver, British Columbia, Canada
[4]School of Health and Related Research (ScHARR), University of Sheffield, Sheffield, UK
[5]University of Bradford, Bradford, UK
[6]College of Health and Medicine, University of Tasmania, Hobart, Tasmania, Australia
[7]Faculty of Health Sciences, University of Hull, Hull, UK
[8]NHS Birmingham and Solihull Clinical Commissioning Group, Birmingham, UK
[9]Primary Care Health Sciences, University of Oxford, Oxford, UK

**Acknowledgements** We acknowledge the contribution from professional and member of the public key stakeholder.

**Contributors** IM conceived and led the project and had overall responsibility for PERISCOPE. All team members developed the embryonic programme theory. AB conducted the searches. AB and GW advised on realist methods. GW also advised on primary care implications and provided methodological oversight. EY screened documents at title, abstract and full text with support from AB and MM. EY, MM and JB coded relevant data into NVivo. EY, MM and JB conducted the data analysis/synthesis. HZ and AH advised on pharmacy aspects and policy implications. TK led PPI. TK and HZ advised on implications in ethnic minorities. All team members contributed to drafting the final report for publication and approved the final draft for submission.

**Funding** Jointly funded by UKRI and NIHR COV0176.

**Disclaimer** The views expressed are those of the author(s) and not necessarily those of the NIHR, UKRI or the Department of Health and Social Care.

**Competing interests** GW is Deputy Chair of the NIHR HTA Prioritisation Committee: Integrated Community Health and Social Care (A) and Member of the NIHR HTA Prioritisation Committee: Integrated Community Health and Social Care (A) Methods Group. AB is a member of the National Institute for Health Research Health Services and Delivery Research Funding Board, the National Institute for Health Research Evidence Synthesis Programme Advisory Group and the National Institute for Health Research School for Social Care Research Commissioning Panel.

**Patient consent for publication** Not required.

**Provenance and peer review** Not commissioned; externally peer reviewed.

**Data availability statement** Data are available upon reasonable request to the lead author.

**ORCID iD**

Ian Maidment http://orcid.org/0000-0003-4152-9704

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
