## [Reviewer comments · BMJ Open]

ARTICLE DETAILS

TITLE (PROVISIONAL)	A Rapid Realist Review of the Role of Community Pharmacy in the Public Health Response to COVID-19
AUTHORS	Maidment, Ian; Young, Emma; MacPhee, Maura; Booth, Andrew; Zaman, Hadar; Breen, Juanita; Hilton, Andrea; Kelly, Tony; Wong, Geoff

VERSION 1 – REVIEW

REVIEWER	Seed, SM Massachusetts College of Pharmacy and Health Sciences
REVIEW RETURNED	10-Mar-2021

GENERAL COMMENTS	Thank you for your submission. The roles of a community pharmacists are rapidly changing. Your methods are well explained and detailed. Your discussion brings up good points- pharmacists and professional pharmacy organizations must advocate for these expanded roles with decision makers. Please review your reference citations- there are many without a DOI number and several are missing the complete citation.
--

REVIEWER	Merks, Piotr Cardinal Stefan Wyszyński University, Faculty of Medicine, Collegium Medicum
REVIEW RETURNED	13-Mar-2021

GENERAL COMMENTS	This is a neatly written manuscript. Unfortunately, paper reflects only to countries with highly advanced pharmacy practice like UK, Canada, USA, Ireland. Additional what was exactly the aim of this paper? In current form it does not have any value for international audience, who would be willing to start pharmacy services improvement in those countries who are behind those mentioned above. Please reflect to foreign practices and link your findings from UK, CA, USA, to other countries, please find if practices has expanded, and in which direction. This manuscript, could potentially be a reference for the rest of countries where our pharmacist colleagues are trying to convince authorities to expand scope of services in community pharmacy. It is missing a larger perspective. Please deliver a new version of this paper. Please kindly describe the problem broader in current form it is very limited.
---

VERSION 1 – AUTHOR RESPONSE

Reviewer: 1

Thank you for your submission. The roles of a community pharmacists are rapidly changing. Your methods are well explained and detailed. Your discussion brings up good points- pharmacists and professional pharmacy organizations must advocate for these expanded roles with decision makers.

Please review your reference citations- there are many without a DOI number and several are missing the complete citation.

Added as requested.

Reviewer: 2

This is a neatly written manuscript. Unfortunately, paper reflects only to countries with highly advanced pharmacy practice like UK, Canada, USA, Ireland.

PERISCOPE was a realist review of secondary published data. Papers were identified using a search strategy that was piloted and refined so that it was fit for purpose. Our initial intention was to search for relevant data from high and middle-income countries (using standard definitions for these). Despite our search strategy (which included grey literature) being developed by a highly experienced information scientist, we did not find relevant data from middle-income countries. An evidence synthesis can only analyse data that are available in the literature. Hence, due to the lack of literature, the results based on the available data, focused on high-income countries. We have indicated in our manuscript the need for more work in this area in a more diverse group of countries.

We have now as requested acknowledged this as a limitation. See page 25, lines 334 to 341: “For this rapid review, given our short timeline of six months, we initially decided to focus on community pharmacy roles and pandemic response in middle to high-income countries. Whilst we identified data from higher-income countries such as Canada, USA, Australia and France, there was very little published from middle-income countries. As such, it is likely that our findings and recommendations are most applicable to high-income countries, although, the lessons learned from high-income countries can serve as an initial reference point for best practice for other countries. The recommendations are “how to’s” for other countries to consider within their own cultural and healthcare settings.”

Additional what was exactly the aim of this paper?

The aim of this study was and is stated in the abstract on page 2. We have now additionally stated the aim in the main text (page 5, lines 91 to 93). “This rapid realist review aimed to understand how community pharmacy can most effectively deliver essential and advanced services, with a focus on vaccination, during the pandemic and in the future.”

In current form it does not have any value for international audience, who would be willing to start pharmacy services improvement in those countries who are behind those mentioned above.

As mentioned above, we have acknowledged this as a limitation and highlighted how the findings can inform practice elsewhere. We are confident that our findings and recommendations are likely to be transferable to other high-income countries as we included documents from such countries. Specifically we included evidence from outside the UK: USA (page 12, line 14: page 13,

line 51; page 16, line 48: page 17, lines 24/25; page 21, lines 23 and 39), Australia (page 16, line 39), France (page 17, lines 24/25), Canada (page 12, line 26; page 13, line 51; page 15, line 44; page 17, line 5; page 21, line 57, page 27, line 55; page 28, line 7) and Ireland (page 15, line 33, page 27, line 55). Moreover, members of the research team were based in Canada and Australia bringing an international perspective.

Please reflect to foreign practices and link your findings from UK, CA, USA, to other countries, please find if practices has expanded, and in which direction. This manuscript, could potentially be a reference for the rest of countries where our pharmacist colleagues are trying to convince authorities to expand scope of services in community pharmacy. It is missing a larger perspective. Please deliver a new version of this paper. Please kindly describe the problem broader in current form it is very limited.

We believe that in the current form as highlighted above, our work is most relevant and applicable to high-income countries. Because of an absence of data from other income countries, we have held back from overclaiming the transferability of our work. In the limitations section, we pointed out this issue (see above). Whilst some of our findings and recommendations may be transferable to low and middle-income countries, to be confident that this is the case more data are needed. Therefore in the Future Research section (page 28, line 407 to 412), we have recommended empirical research in countries where there is little or no published literature. Such research would go some way to address the important issue raised by the peer-reviewer.

We look forward to hearing from you at your earliest convenience.

VERSION 2 – REVIEW

REVIEWER	Merks, Piotr Cardinal Stefan Wyszyński University, Faculty of Medicine, Collegium Medicum
REVIEW RETURNED	10-Apr-2021

GENERAL COMMENTS	This a very well written paper. I would only add a bit more in the introduction section about the legal extension of work of pharmacists in other countries during COVID 19 pandemic. It is also worth to show a global aspect. We are all fighting and supporting the system. It would good to have a chance to refer to this paper even if a manuscript is country specific. When the manuscript present system solution or shows how it shall be done by the book, it present an opportunity for foreign pharmacists to use it in local negotiations with stakeholders or design a similar project. Please look and add few more references from other countries how system gain by involving pharmacist in many new roles.
--

VERSION 2 – AUTHOR RESPONSE

Reviewer 2:

This a very well written paper. I would only add a bit more in the introduction section about the legal extension of work of pharmacists in other countries during COVID 19 pandemic. It is also worth to

show a global aspect. We are all fighting and supporting the system. It would good to have a chance to refer to this paper even if a manuscript is country specific. When the manuscript present system solution or shows how it shall be done by the book, it present an opportunity for foreign pharmacists to use it in local negotiations with stakeholders or design a similar project. Please look and add few more references from other countries how system gain by involving pharmacist in many new roles.

Thank you for your comments. We did briefly discuss application in other countries in the discussion, but we agree with you that this could be expanded. We have therefore expanded this section with appropriate references (please see lines 340 to 345). As part of this expansion, we cover legislative changes, as requested. We feel that this amendment sits better in the discussion e.g. it is about our results in the wider context.

We look forward to hearing from you at your earliest convenience.

VERSION 3 – REVIEW

REVIEWER	Merks, Piotr Cardinal Stefan Wyszyński University, Faculty of Medicine, Collegium Medicum
REVIEW RETURNED	21-Apr-2021
GENERAL COMMENTS	Thank you so much for addressing all my comments. I have no more comments to be added.